# Measuring how motivation affects information quality assessment: A gamification approach

**Marko Poženel, Aljaž Zrnec, Dejan Lavbič** *

University of Ljubljana, Faculty of Computer and Information Science, Ljubljana, Slovenia

* Dejan.Lavbic@fri.uni-lj.si

## Abstract

### Purpose

Existing research on the measurability of information quality (IQ) has delivered poor results and demonstrated low inter-rater agreement measured by Intra-Class Correlation (ICC) in evaluating IQ dimensions. Low ICC could result in a questionable interpretation of IQ. The purpose of this paper is to analyse whether assessors' motivation can facilitate ICC.

### Methodology

To acquire the participants' views of IQ, we designed a survey as a gamified process. Additionally, we selected Web study to reach a broader audience. We increased the validity of the research by including a diverse set of participants (i.e. individuals with different education, demographic and social backgrounds).

### Findings

The study results indicate that motivation improved the ICC of IQ on average by 0.27, demonstrating an increase in measurability from poor (0.29) to moderate (0.56). The results reveal a positive correlation between motivation level and ICC, with a significant overall increase in ICC relative to previous studies. The research also identified trends in ICC for different dimensions of IQ with the best results achieved for completeness and accuracy.

### Practical implications

The work has important practical implications for future IQ research and suggests valuable guidelines. The results of this study imply that considering raters' motivation improves the measurability of IQ substantially.

### Originality

Previous studies addressed ICC in IQ dimension evaluation. However, assessors' motivation has been neglected. This study investigates the impact of assessors' motivation on the measurability of IQ. Compared to the results in related work, the level of agreement achieved with the most motivated group of participants was superior.

**Data Availability Statement:** The data used in the paper is publicly available in the GitHub repository https://github.com/DejanL/PlosOne-IQ-data.

**Funding:** The author(s) received no specific funding for this work.

**Competing interests:** The authors have declared that no competing interests exist.

# 1 Introduction

Making the best possible decisions requires information of the highest quality. As the amount of information available grows, it becomes increasingly difficult to distinguish quality from questionable information [1]. The problem of poor information quality can weaken our decision processes, so we need more reliable measures and new techniques to assess the quality of information [2–4]. Unfortunately, such assessment can itself be very demanding [1, 2, 5].

In general, the term *information quality (IQ)* represents the value of information for a given usage. However, IQ often refers to people's subjective judgment of the goodness and usefulness of information in certain information use settings [5, 6]. The literature has widely adopted a multidimensional view of IQ [1, 7] to support more effortless management of its complexity.

The measurability of IQ has gained substantial attention in recent years [5, 8, 9]. Most research in this field has been limited to measuring the quality of structured data (e.g. data in databases where a scheme is defined in advance) [10–12]. Measuring the IQ of unstructured data (e.g. Wikipedia articles) requires different approaches that include interdisciplinary components [13]. The research community proposed several determinants of IQ and there is a growing concern regarding how to best identify quality information [1]. Only a few studies presented inter-rater agreement results using Interclass Correlation Coefficient (ICC) statistics, and multiple guidelines for the interpretation of ICC inter-rater agreement values exist [14–16]. Regardless of the ICC interpretation used, the values reported in recent studies are poor or at best moderate [2, 4]. This demonstrates that reaching consensus among various raters is difficult when measuring IQ.

Research problems regarding efficient IQ measurement remain relatively underexplored. Previous papers studied some of the cues that affect IQ assessment on selected sources of data [1, 2]. However, the research community needs additional case studies to evaluate the inter-rater reliability of IQ dimensions (a single aspect of data that can be measured and improved) in various settings to help increase the external validity of the cues and factors.

Being motivated means having an incentive to do something [17]. Intrinsically motivated does something for its own sake, for the sheer enjoyment of a task, while extrinsically motivatied does something in order to attain some external goal or meet some externally imposed constraint [18]. To the best of our knowledge, no previous research has investigated how motivation affects IQ assessment and whether it has a significant impact on inter-rater agreement. In this paper, we propose a new approach that improves the measurability of IQ by considering various IQ dimensions. Specifically, we study the effect of motivation on IQ measurement and inter-rater reliability. Researchers have always seen motivation as an important factor that influences learning performance [19–21]. Our goal in this work is to corroborate that motivation also affects the measurability of IQ.

In related work, [2] studied the measurability of IQ in Wikipedia articles, and [4] narrowed the object of a study to individual paragraphs. In this work, we evaluate IQ of hints that (i) correspond to selected IQ dimensions, (ii) have diversified predefined quality, and (iii) help participants in progressing through the gamified process. Specifically, we evaluate the relevance of gamified task hints targeting IQ dimensions of accuracy, objectivity, completeness, and representation. We are interested in the consistency between multiple raters assessing the same set of hints in a hands-on assignment.

This study contributes to the existing literature concerning IQ measurability and inter-rater reliability. It extends the work presented in [1, 2]. To support comparison, we use the categorization of IQ dimensions defined by [22], previously used in similar studies [1, 2, 4].

The remainder of the paper is structured as follows. In section 2, we review related work and introduce the problem statement and our proposed solution. We follow this with

presentation of the empirical study design and the experiment in section 3. Then we present the results and discuss the implications and limitations in section 4. Finally, in section 5, we present our conclusions, limitations and suggest directions for future work.

## 2 Related work

### 2.1 Assessing the quality of information

With the growing amount of information published every day, IQ has gained huge social importance [11, 23–25]. Several studies stressed the increase in interest in IQ [2, 5, 26]. This body of research has often focused on dimensions of IQ and what factors affect its measurability [1, 2]. Fewer studies focused on the measurability and assessment of IQ. However, issues with IQ are becoming growingly prevalent [22], especially with the rise of user-generated content (e.g. Wikipedia) and citizen science, where users participate in simultaneously creating and editing information. Poor-quality user-generated content (UGC) can present an issue for information retrieval services [27] and individuals.

The community considers IQ assessment demanding because sources of information lack metadata and IQ criteria are often subjective [5]; which makes it hard for multiple raters to agree upon an object's IQ [2]. Assessment of an object's (Wikipedia article, paragraph, hint, etc.) IQ depends on several factors, including object itself, and the assessor's prior knowledge, differences in domain expertise, cognitive or demographic traits. Previous research that has studied IQ assessment agrees that IQ is not a uniform construct and that it consists of multiple dimensions [1, 2, 22]. Thus, we cannot assess IQ as a whole but according to its underlying dimensions. The research community has proposed several frameworks and underlying dimensions for the assessment of IQ. [28] defined a set of dimensions and a framework where dimensions are grouped into a hierarchical model of IQ aspects and their criteria. Several authors [22, 29, 30] later investigated the initial set of dimensions defined by [28] and evaluated the degree to which individual dimensions comply with the needs or expectations of users [26]. In this paper, we apply the set of quality dimensions (accuracy, completeness, objectivity and representation) that researchers used in most previous empirical studies [1, 7].

The dimensions are defined as follows: Accuracy indicates factual correctness of the data and absence of errors (incorrect information, references to non-authoritative sources, and spelling errors); Completeness refers to sufficient coverage of information appropriate for an encyclopedic entry and to the lack of omission of relevant facts (e.g., missing introductory and background information that would help explain the topic's relevance, importance, or its history); Objectivity pertains to an impartial view of the topic and to the absence of subjective language, opinions stated as facts, the omission of alternative perspectives or existing controversies, or a deliberate misrepresentation; and Representation refers to clarity and ease of understanding at a readership level accessible to the general public (using diagrams when required), rational organization, consistent presentation using a single "voice", and concise formatting.

[2] focused on IQ dimensions and the extent of agreement (i.e. inter-rater reliability) that could be achieved when rating the aforementioned four IQ dimensions. They found that some IQ dimensions are more difficult to assess than others and noted that assessors often employ heuristics during IQ assessment. [1] explored the role of heuristic principles in IQ assessment, investigating how the consistent application of heuristic principles affects inter-rater agreement according to IQ dimensions. [26] investigated the effects of satisfaction and complexity on the perception of IQ dimensions and found that satisfied users place a higher weight on qualitative than quantitative aspects of IQ. [31] focused on cognitive heuristics in credibility

evaluation, studying the heuristics that information consumers use when deciding what sources and information to trust online.

Several research efforts sought to assess the IQ of content in a collaborative-writing environment, UGC, and citizen science. [5] examined how non-expert users evaluate the quality of Hebrew Wikipedia contents with a focus on identifying the cues and criteria that users find helpful to assess the quality of Wikipedia articles. [32] proposed data derived from UGC and citizen science be used for studying innovative approaches to IQ management. [4] provide informed insights on students' perception of IQ. They proposed an approach to improve the relevance of Wikipedia articles to meet students' needs.

All of these studies highlighted issues with IQ assessment. With these issues in mind, our study focuses on factors that may have positive effects on reliable measurement and facilitate the assessment of IQ.

## 2.2 Gamification and motivation

In the previous work that studies the measurability of IQ, authors achieved mediocre results [1, 2]. However, measuring IQ depends on elusive factors and presents a challenging task [2]. The measures for evaluating IQ depend on the source and also the criteria may not be viewed equally by the users and researchers [2]. Assessment of quality depends on the "fitness" of the data to one's specific assessment purposes [1]. We assume that the assessors' motivation can also be a key factor for quality assessment of IQ. [33] showed that motivation is a crucial success factor, especially in learning. [34] studied the influence of various types of motivations on employees' knowledge sharing behaviors and found that hard reward is a key motivational factor next to soft reward. Motivated assessors may also contribute to a higher inter-rater reliability. To improve assessors' motivation, we introduce the concept of gamification in the assessment process.

Gamification (Gameful design) [35] is a concept where we use game-like elements in various systems to increase user participation, motivation, improve engagement, or to retain users continue using the system. In the literature, gamification is often defined as the use of game design elements in non-game contexts [36]. It is an innovative approach to stimulate motivation [37]. Motivation is hardly unitary phenomenon [17], and can be studied from different perspectives. In Self-Determination Theory (SDT) [38], we distinguish between different types of motivation based on the different reasons or goals that initiated an action [17]. The most basic distinction is between intrinsic motivation and extrinsic motivation [17]. Intrinsic motivation is defined as the wish or tendency to execute an action for its own sake, for example because of its interesting, challenging or exiting nature [38]. It enables high-quality learning and creativity [17]. Extrinsic motivation contrasts with intrinsic motivation, and refers to the pursuit of an instrumental goal, i.e to achieve results that are not related to action performed [17, 39]. Addressing an individual's intrinsic motivation to play and have fun, we can also define gamification as the concept of leveraging the psychological predisposition to engage in gaming, using mechanisms that game designers applied in making video games, as a potential means to make real-world activities more engaging [40]. Gamification proved to be successful in addressing individuals' motivation and increasing the user's engagement. SDT assumes three universal psychological needs for competence, autonomy, and social relatedness [41]. The fulfilment of these needs is especially relevant for fostering intrinsic motivation [37]. Also the integration of extrinsic motivation can be addresses by fulfilling these needs as well [37, 42]. According to SDT, players are likely to be motivated if they experience the feeling of competence, autonomy and social relatedness [37]. In their study of gamification in the workplace, [43] found that if extrinsic motivation is internalized, it can support needs satisfaction,

intrinsic motivation, and behavioral intention. Intrinsic motivation was positively associated with behavioural intention in workplace gamification use [43]. Gamification is used in application fields like sports [44], health [45], sustainability [46], education [47], marketing and business [48]. [49] reported improved learning using computer games in training applications. In e-learning applications, we can use gamification to enhance motivation. In companies, we can employ gamification to increase employee engagement and to motivate them to perform their tasks with more enthusiasm [50]. Even when the introduction of gamification into training did not prove to increase outcomes, it increased the levels of learner motivation to acquire those skills [51]. Gamification domain is vast and the research community continue to discover new areas of application [43].

The main components of gamification are game elements, which denote specific game components that can be used in gamification [37, 52]. Game elements such as points, levels and leaderboards have become a constant in gamification, especially due to their use in games [53]. The relationship between game elements and Self-Determination Theory is presented in [52]. [37] also analyzed game elements and linked them to motivational mechanisms that they primarily refer to. In the literature, it has been argued that thoughtful implementation of game elements may improve intrinsic motivation by satisfying users' innate psychological needs [53–56]. [53] studied the effects of individual game elements on motivation and performance. They found that gamification did not affect intrinsic motivation, but their results suggest that in the given context game elements acted as extrinsic incentives. However, [57] stressed that gamification is not only the addition of game elements and game design to non-game processes but rather the development and design processes supported by extant research. Gamification studies how we can motivate users and change the process that it gamifies [57, 58].

To our knowledge, previously, gamification has not been used to improve IQ assessment and inter-rater agreement. The existing literature that investigated IQ assessment builds on the well-established classic non-gamified process. Existing assessment processes build on extrinsic motivation to perform tasks-at-hand. In this study, we introduce a novel IQ assessment process that employs gamification. We analyzed the existing assessment process, shortened the length of the source document under assessment, and created a new gamified assessment application. The final application contains game-like components, but it has a functional non-game purpose and elements, which are not game-like [59]. Game elements include points, levels, choice elements, progress bars, and leaderboards. Our goal was to increase the assessor's intrinsic motivation to play and have fun and to positively impact the assessors' attitude to perform the IQ assessment tasks. Besides, we included extrinsic motivation through rewards to receive acknowledgment in the hall of fame scoreboard. Thus, we hypothesize a positive effect of gamification on motivation and finally on IQ assessment score.

## 2.3 Problem statement and proposed solution

The existing work primarily focuses on heuristic principles (i.e. cognitive decision-making processes) that help assessors with IQ assessment and the effects of consistent application of heuristic principles on the inter-rater reliability. However, fewer studies focused on the cues that influence IQ assessment and inter-rater reliability. Our study contributes to a better knowledge of cues that affect inter-rater agreement levels when assessing IQ. It primarily focuses on the effects of motivation on inter-rater reliability, while it also investigates differences in the measurability of IQ dimensions and their corresponding inter-rater agreements.

Assessment is a complex and mentally labor-intensive task. We believe that the degree of effort that assessors are willing to put into the IQ assessment process affects its result, and

hence, inter-rater reliability. If all raters are highly motivated, the difference between their assessments will decrease, and inter-rater agreement will improve.

To motivate participants to do their best at assessing IQ dimensions, we introduced gamification to the assessment process. We adjusted the gamified content to provide an engaging assessment environment that supports assessors in their assessment process [58]. Intrinsically motivated activities facilitate assessors to perform tasks without any kind of conditioning [60], while elements of extrinsic motivation help to perform work tasks through rewards [43, 61].

Focusing on motivation, the researchers conducted some research in the field of IQ. However, they have not thoroughly studied the impact of motivation on assessment consistency, nor did they employ the concept of gamification. Previous research has been more interested in how increased motivation affects the more consistent use of heuristics, resulting in a possible higher inter-rater reliability. For instance, [31] focused on the use of cognitive heuristics in credibility evaluation in an online environment.

In the literature, [5] indirectly acknowledged the importance of motivation when using the Elaboration Likelihood Model (ELM) to explain why article length is often viewed as an indicator of quality. According to the ELM, users approach the problem of evaluation systematically when they are motivated and have the knowledge about the relevant topics, but make use of rules of thumb when their motivation and relevant knowledge are lacking.

Concentrating on the source, previous approaches in the literature mainly focused on assessing Wikipedia articles; to our knowledge, few or no studies focused on blogs and other sources. [1] indicated that the measurability of IQ depends on media type and task context.

Based on our review of the literature and study design, we formulate the following research questions (RQ):

- RQ1: To what extent can motivation affect the measurability of IQ for short hints used in a gamified process?

- RQ2: How does motivation influence individual IQ dimensions in terms of inter-rater agreement in the assessment of IQ?

## 3 Method

### 3.1 Evaluation mechanics

To address the research questions, we conducted an online quantitative case study. Participation was voluntary and user consent was obtained before the start. As presented in detail in section 3.2, we included a diverse set of participants (i.e. individuals with different education, demographic and social backgrounds), which amplified the validity of our research. To reach a broader audience, we selected a Web study, and to acquire the participants' views of IQ, we designed a survey in the form of a gamified process.

Previous studies on measuring IQ dimensions focused primarily on students [2] and university librarians [1]. The main drawback of existing studies is the small group of participants. In the present study, we followed a design that would investigate smaller data sources under study but use existing data dimensions and existing estimation metrics with a much bigger set of participants as further discussed in section 3.2.

We employed a gamification principle to measure the influence of motivation on the evaluation of IQ. For our experiment, we developed a tool in the form of a Web-based gamified software tool. The overall gamified purpose of the assessment application is to save and raise a little bird to adulthood and return it to the wild (see details in section 3.3). The objective is to complete the gamified process with a minimum number of attempts to receive more points,

which addresses the motivational aspect. Because of the dynamic conditions under which the participants gain points, more motivated participants collect a higher number of points for their effort. For higher user engagement in the gamified process of measuring IQ, we included the following game elements: leaderboard (visual display of social comparison), levels (player's progressive) and points (virtual rewards against the player effort) as those elements improve the motivation and performance of participants [53].

The research goal of the gamified process is to assess hints' IQ that corresponds to selected IQ dimensions and help participants in progressing through the gamified process. We argue that the participants' success in resolving the gamified task strongly correlates to the consistency of given IQ evaluations of participants in selected dimensions.

## 3.2 Participants

Our study had a total of 1225 participants who participated from April 2015 to March 2020. Initially, we directly targeted undergraduate University students whom we had direct access and then all potential participants by utilising mail and social media campaigns. In the process of data cleaning, we excluded participants that did not answer all 24 questions (4 game levels with 6 questions each) and spent a total time of less than 5 min (quick random selection of responses) or more than 3 h (multiple breaks while playing) to complete the gamified process.

We were then left with 1062 responses with the median time to complete all 4 game levels of 11 min 50 s. There were 30.7% female and 69.3% male, with ages ranging from 15 to 69 with a median value of 20 years old.

In general, we targeted a population that has finished high school, since they have more experience with poor IQ. A total of 40.5% of participants stated that poor IQ deeply disturbs them and 37.5% stated that they at least bother about poor IQ. We sought to increase the diversity of participants to enhance the external validity of the research; the participants were 57.6% undergraduate students and 42.4% non-students.

Although, the study required no prior knowledge to participate, we included all participants in a pre-training that provided an introduction to the IQ dimensions (completeness, accuracy, representation, and objectivity) that they had to evaluate later in the gamified process. Before performing gamified tasks, they also completed an evaluation task in which they were asked to measure these IQ dimensions in a short paragraph.

## 3.3 Measuring IQ dimensions

The gamified process consists of four levels. At each game level, we evaluate one of the selected IQ dimensions (completeness, accuracy, representation, and objectivity) highlighted in section 2. Fig 1 depicts an overview of the measurement of IQ dimensions, while Fig 2 shows comprehensive details. For every IQ dimension, we presented evaluating objects in a random order (see activity A2 in Fig 1). Each object is associated with a hint of a different predefined level of correctness (see Eq (5)). The rater then evaluates the IQ of a hint (for finding an object within a gamified process) before employing it (see activity A6 in Fig 1) to find the correct object. The number of points awarded is a function of the number of attempts and the level of correctness of a given hint (see Eq (6)). Once the rater finds the correct object, he evaluates again the IQ of the same hint, where a calibration to prior evaluation is possible (see activity A9 in Fig 1). The gamified proces ends when the rater finds all objects within a given IQ dimension and evaluates IQ dimensions.

Fig 2 depicts that the first step in evaluating IQ dimension is displaying the game rules for the $g$-th level (see activity A1 in Fig 2), where $g \in [1, 4]$. At the start, each player receives general information about the story of the level and tasks that he must accomplish, along with

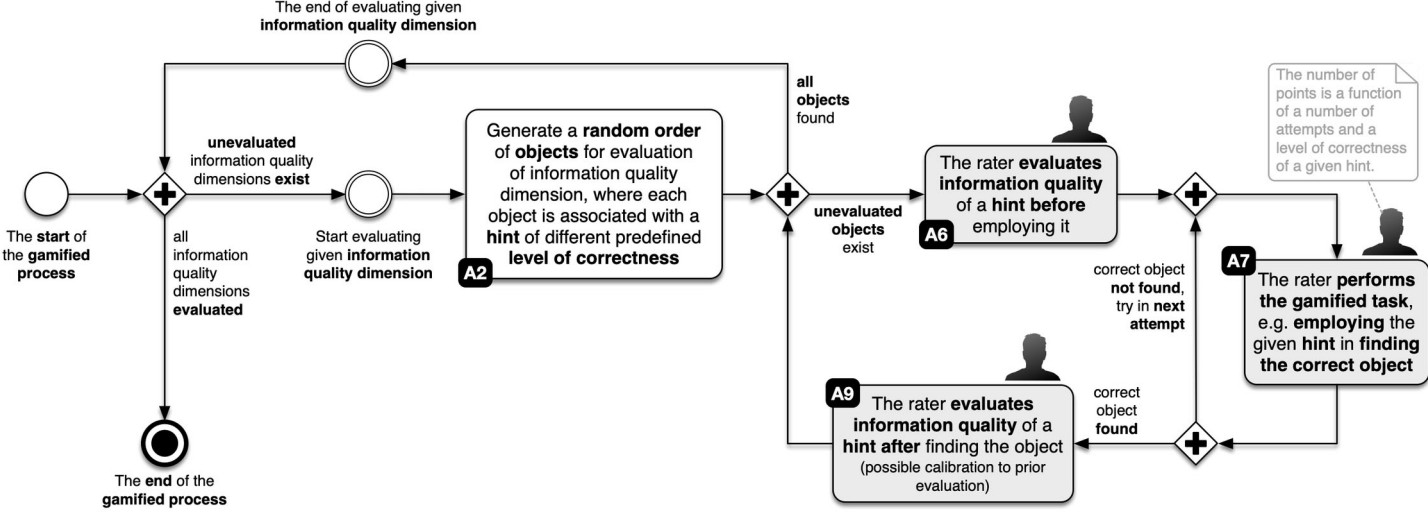

**Fig 1. Overview of measuring selected IQ dimensions.**

detailed instructions. At all times there is a progress bar available at the top of the screen that informs the player of his status within the gamified process.

To successfully finish each level of the gamified process, participants must find the correct object $o(g)$, based on the gamified task's context and the hint provided. The objects for each of the game's levels $o(g)$ are selected and displayed to the participant in a random order (see activity A2 in Fig 2).

The **first level** focuses on **completeness**; participants must find a hidden object from the following set

$$o(1) = \{\text{birdie}, \text{worms}, \text{fox}, \text{strawberries}, \text{key}, \text{treasure chest}\} \tag{1}$$

Presented hints are of various completeness levels, where the most complete hint provides information for unique identification of a location of a hidden object, while the least complete hint involves a great level of ambiguity (e.g. there are several possible locations of a hidden object).

The **second level** focuses on **accuracy**; participants must weigh food and select the correct weight of the following objects:

$$
\begin{aligned}
o(2) = \{ \\
&\{\text{worms} + 2\ \text{flies} + \text{mosquito} + \text{blackberries}, \\
&\{\text{crumbs} + \text{worms} + 6\ \text{bugs}\}, \\
&\{\text{crumbs} + 6\ \text{flies} + 2\ \text{bugs}\}, \\
&\{\text{crumbs} + \text{bug} + \text{blackberries}\}, \\
&\{2\ \text{crumbs} + \text{worms}\}, \\
&\{2\ \text{crumbs} + 4\ \text{mosquitos} + 3\ \text{flies} + 2\ \text{bugs}\} \\
\}
\end{aligned}
\tag{2}
$$

Presented hints are associated with scales of various accuracy, from the most accurate with the exact measurement and the least accurate with the false range of measurement.

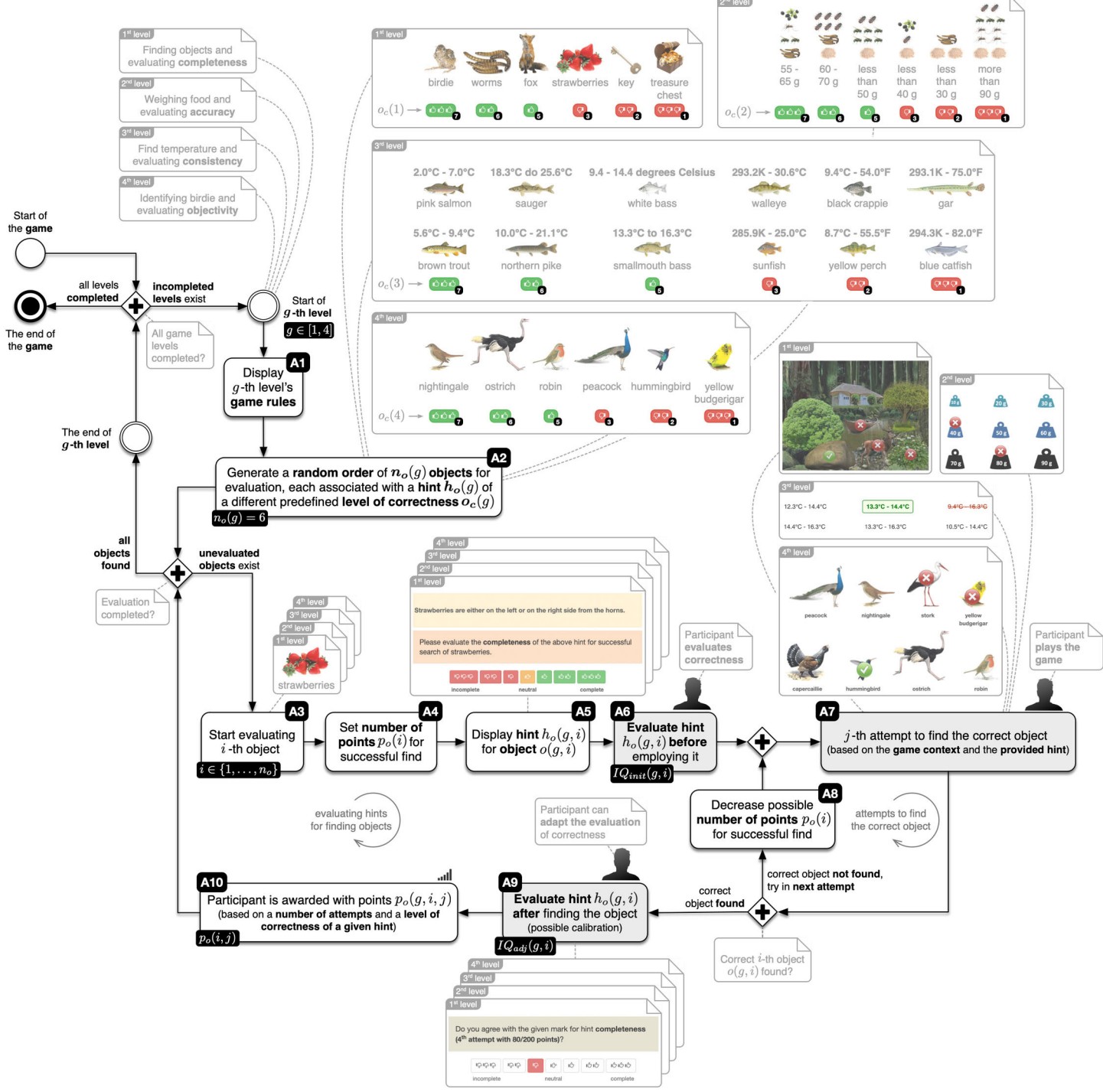

**Fig 2. Detailed process of measuring selected IQ dimensions.**

The **third level** focuses on **representation**; participants must find the correct temperature range for a living habitat of the following fish:

$$o(3) = \{$$

$$\{\text{brown trout and pink salmon,}$$
$$\{\text{sauger and northern pike}\},$$
$$\{\text{white bass and smallmouth bass}\},$$
$$\{\text{walleye and sunfish}\},$$
$$\{\text{black crappie and yellow perch}\},$$
$$\{\text{gar and blue catfish}\}$$

$$\}$$

(3)

Presented hints are associated with the representation of temperature ranges that are available in various units (Celsius, Fahrenheit, and Kelvin), where the most consistent representation includes a range with only one unit of measurement, while the least consistent representation presents a mix of various units.

The **fourth level** focuses on **objectivity**; participants must indicate the correct bird

$$o(4) = \{$$

$$\text{nightingale, ostrich, robin, peacock,}$$

$$\text{hummingbird, yellow budgerigar}$$

$$\}$$

(4)

Presented hints include bird's origin, size, and color that are presented by various levels of objectivity with different people, from the most objective ornithologists to the least objective bankers, programmers, and wall painters.

There are $n_o(g) = $ **6 objects** at every level $g$, each associated with a hint $h_o(g)$ of different **level of correctness** $o_c(g) = \{1, 2, 3, 5, 6, 7\}$, where $o_c(g) = \{1, 2, 3\}$ is associated with correct hints (with value 1 the most correct hint) and $o_c = \{5, 6, 7\}$ are incorrect hints (with value 7 the most incorrect hint). The main part of the task is to evaluate the $i$-th object based on the provided hint (see activity **A6** in Fig 2).

The function of correctness $o_c(g)$ differs for every level $g$ and represents the selected IQ dimension being measured

$$o_c(g) = \begin{cases} o_c = \text{completeness} & ; \quad \text{if } g = 1 \\ o_c = \text{accuracy} & ; \quad \text{if } g = 2 \\ o_c = \text{representation} & ; \quad \text{if } g = 3 \\ o_c = \text{objectivity} & ; \quad \text{if } g = 4 \end{cases}$$

(5)

At every level of the game $g$, participants receive $n_o(g) = 6$ objects in random order for evaluation. When searching for the correct object, participants can try and find the correct object if their previous attempt was incorrect (see activity **A7** in Fig 2).

The **points** awarded for a correct answer $p_o(i,j)$ are decreasing linearly with the **number of attempts** $j \geq 1$ and a predefined **level of correctness** $o_c$ (the less correct the hint is, the more points are awarded if the object is correctly identified) as depicted in Eq (6). We based the

quality of hints on a previous analysis and rules that pertain to the dimension under study.

$$p_o(i,j) = \begin{cases} 10 \cdot (6-j) \cdot (7-o_c) & ; \quad \text{if } j < 6 \text{ and } o_c(i) \in \{1,2,3\} \\ 10 \cdot (6-j) \cdot (8-o_c) & ; \quad \text{if } j < 6 \text{ and } o_c(i) \in \{5,6,7\} \\ 0 & ; \quad \text{otherwise} \end{cases} \tag{6}$$

Each participant can make multiple attempts to find the object (see activity **A7** in Fig 2) but is, according to Eq (6), motivated to find the correct answer in the minimum number of attempts. Each failed attempt reduces the number of points awarded (see activity **A8** in Fig 2) at a given level $g$ and consequently in the game as a whole.

When a participant at a given game level $g$ starts evaluating the $i$-th object $o(g, i)$ (see activity **A3** in Fig 2) on a 7-point Likert scale, an associated hint $h_o(g, i)$ with a predefined level of correctness $o_c(g, i)$ is displayed (see activity **A5** in Fig 2). The hint is evaluated with $IQ_{init}(g, i)$ (see activity **A6** in Fig 2) before the participant tries to find the correct object $o(g, i)$ (see activity **A7** in Fig 2) in a minimal number of attempts $j$, because points $p_o(g, i, j)$ are associated with the number of attempts required for success. After the correct object $o(g, i)$ is found, the previous evaluation $IQ_{init}(g, i)$ can be calibrated with $IQ_{adj}(g, i)$ (see activity **A9** in Fig 2); the participant can alter previously given scores for an IQ dimension, if desired.

The evaluation process of IQ dimensions is complete at the end of the 4-*th* level. At the end, the system gives the participant a score and his overall position in the rankings.

## 3.4 Evaluation metrics

The interclass correlation coefficient is a reliability index widely used for intra-rater and inter-rater reliability analyses. Since we measured the variation between raters measuring the same group of objects in this work, we focused on inter-rater reliability. According to the guidelines, proposed by [16], we select ICC(2,1) as a measure of agreement for our inter-rater reliability study (see Fig 3 and Table 1).

The attributes of the ICC(2,1) are:

- the model is two-way random with k raters randomly selected and each hint (total of n hints) measured by the same set of k raters,

- the number of measurements is single measures and reliability is applied to a context where a single measure of a single rater is performed,

- the metric is absolute agreement, where the agreement between raters is of interest, including systematic errors of both raters and random residual errors.

ICC(2,1) is defined as follows

$$ICC(\mathbf{2,1}) = \frac{BMS - EMS}{BMS + (k-1) \cdot EMS + \dfrac{k}{n} \cdot (JMS - EMS)} \tag{7}$$

where

- $\mathbf{WMS} = \frac{WSS - BSS}{n \cdot (k-1)}$ is **Within Mean Squares** (from one-way ANOVA),

- $\mathbf{BMS} = \frac{BSS}{n-1}$ is **Between Objects Mean Squares** (from one-way ANOVA),

- $\mathbf{JMS} = \frac{JSS}{k-1}$ is **Joint (between raters) Mean Squares** (from two-way ANOVA),

- $\mathbf{EMS} = \frac{ESS}{(n-1) \cdot (k-1)}$ is **Error (residual) Mean Squares** (from two-way ANOVA).

- **ESS** = $WSS - BSS - JSS$ is **Error (residual) Sum of Square** (from two-way ANOVA),

- **JSS** = $n \cdot \sum_{j=1}^{k} \left( \frac{1}{n} \cdot \sum_{i=1}^{n} v_{ij} - m_a \right)^2$ is **Joint (between raters) Sum of Square** (from two-way ANOVA),

- **BSS** = $k \cdot \sum_{i=1}^{n} \left( \frac{1}{k} \cdot \sum_{j=1}^{k} v_{ij} - m_a \right)^2$ is **Between Objects Sum of Squares** (from one-way ANOVA),

- **WSS** = $\sum_{i=1}^{n} \sum_{j=1}^{k} (v_{ij} - m_a)^2$ is **Within Sum of Squares of all raters** (from one-way ANOVA) and

- **$m_a$** = $\frac{1}{n \cdot k} \sum_{i=1}^{n} \sum_{j=1}^{k} v_{ij}$.

To measure the reliability of scale we also calculate Cronbach's alpha ICC(3,k), which is defined as

$$\alpha = \frac{BMS - EMS}{BMS} \tag{8}$$

## 4 Results and discussion

### 4.1 Results

Participants in our research evaluated the IQ dimensions of hints in a gamified process, where we rewarded their effort with a score. Table 1, Figs 3 and 4 show the results referring to both research questions. The experiment had a two-fold purpose. First, to measure the inter-rater reliability agreement as ICC(2,1) in evaluating various IQ dimensions. Second, to measure the motivation of participants in the gamified process. The involvment of the participants in a form of motivation is measured by game points, related to the performance of players further determined by the number of attempts and predefined level of correctness of a given IQ dimension. The mechanics of points calculation are defined in Eq (6), where each aprticipant can make multiple attempts to find the object but is motivated to find the correct answer in the minimum number of attempts. Each failed attempt reduces the number of points awarded to the player at a given level and consequently in the game as a whole. The aforementioned results ICC(2,1) and groups Q are not related in terms that ICC(2,1) focuses on agreement with other raters, while groups focus on the provided value of IQ dimension by the rater in relation to the predefined level of correctness associated with the dimension.

Table 1 presents detailed inter-class agreement results ICC(2,1) (denoted by ICC) including the measured reliability of scale ICC(3,k) (denoted by $\alpha$) for various constructs regarding different groups of participants. We divided the participating players' scores into four groups according to the number of points scored in the gamified environment. The groups in Fig 3 are arranged in ascending order by quartile. Group Q4 represents those who achieved mediocre results, while Q1 represents the highest-scoring players.

There are five IQ dimension groups; four for every dimension under investigation, and CIQ, the mean value of all four dimensions.

Fig 3 depicts ICC results for the selected set of IQ dimensions (completeness, accuracy, representation, and objectivity) for different participants' groups (Q1, Q2, Q3, and Q4).

There are multiple guidelines for the interpretation of ICC inter-rater agreement values [14–16]. Fig 4 summarizes the results of our study by incorporating all aforementioned ICC interpretations. We can observe that highly motivated participants achieved substantially better results compared to the unmotivated ones regardless of the interpretation chosen.

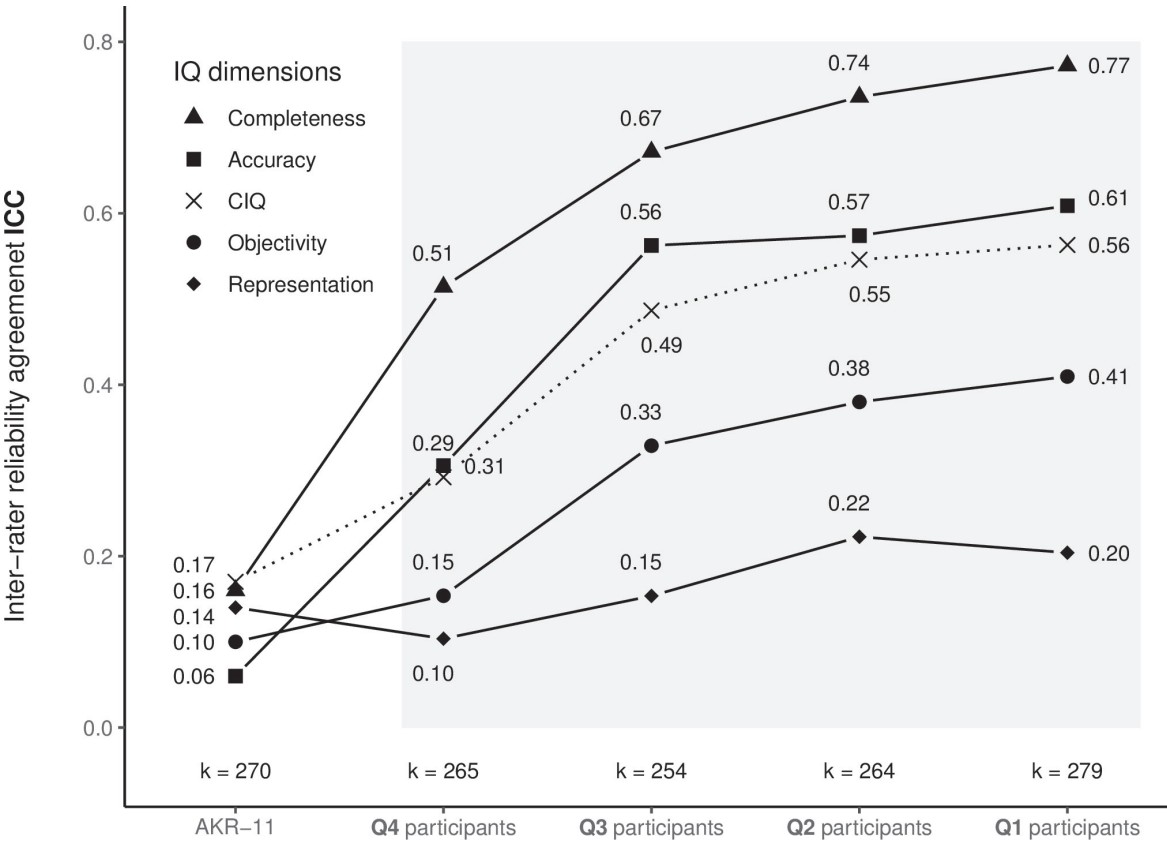

**Fig 3. Interclass correlation vs. performance of players.** The y-axis represents the ICC, while the x-axis portrays four groups of participants. The groups are divided into quartiles according to the points that participants scored when performing the gamified process. The first column AKR-11 contains results of a similar IQ study, in which [2] studied the measurability of IQ on the same set of IQ dimensions used in our study. The four highlighted columns (rightmost) exhibit the results of our study, where each column represents one group of players.

## 4.2 Discussion

When analyzing the differences in ICC between the various groups, we can observe that the agreement level increases with players' performance in terms of the final score achieved (see Fig 3). Raters who were more motivated more carefully rated the hints by a given dimension, which lead to more homogenous assessments. Group Q4 with members that attained mediocre results in the gamified process, represents participants who had the poorest motivation and did not focus on the task-at-hand. The results for this group represent a foundation, a basic ICC with which to compare other group' who were more motivated.

When comparing the other three quartiles (Q3, Q2, and Q1), it is evident that inter-rater agreement for all dimensions increases consistently with the increasing game score. We can observe the same for the average score, CIQ. The exception to the rule is the dimension representation; Q1 achieved a slightly lower inter-rater reliability than Q2. However, the ICC for representation is still higher than for Q4 and Q3.

There is a substantial increase in ICC between groups Q4 and Q3 for all dimensions. The results indicate that participants who were even slightly motivated quickly achieved better ICC. Therefore, for RQ1, we can conclude that increased motivation reinforces the measurability of IQ for short hints in the context of a gamified environment.

**Table 1. ICC results in our research.**

| IQ dimension | Q | n | k | BMS | WMS | JMS | EMS | ICC(2,1) | ICC(3,k) |
|---|---|---|---|---|---|---|---|---|---|
| Completeness | Q4 | 6 | 265 | 871.10 | 3.10 | 6.36 | 2.45 | 0.51 | 1.00 |
| Completeness | Q3 | 6 | 254 | 1,202.42 | 2.31 | 3.75 | 2.02 | 0.67 | 1.00 |
| Completeness | Q2 | 6 | 264 | 1,357.12 | 1.84 | 3.06 | 1.60 | 0.74 | 1.00 |
| Completeness | Q1 | 6 | 279 | 1,494.92 | 1.58 | 2.54 | 1.39 | 0.77 | 1.00 |
| Accuracy | Q4 | 6 | 265 | 442.82 | 3.77 | 9.83 | 2.56 | 0.31 | 0.99 |
| Accuracy | Q3 | 6 | 254 | 764.38 | 2.34 | 5.05 | 1.79 | 0.56 | 1.00 |
| Accuracy | Q2 | 6 | 264 | 841.57 | 2.36 | 5.49 | 1.74 | 0.57 | 1.00 |
| Accuracy | Q1 | 6 | 279 | 956.64 | 2.20 | 4.68 | 1.70 | 0.61 | 1.00 |
| Representation | Q4 | 6 | 265 | 137.10 | 4.39 | 12.89 | 2.69 | 0.10 | 0.98 |
| Representation | Q3 | 6 | 254 | 161.69 | 3.46 | 9.37 | 2.28 | 0.15 | 0.99 |
| Representation | Q2 | 6 | 264 | 229.97 | 3.02 | 8.49 | 1.92 | 0.22 | 0.99 |
| Representation | Q1 | 6 | 279 | 227.29 | 3.15 | 10.07 | 1.77 | 0.20 | 0.99 |
| Objectivity | Q4 | 6 | 265 | 165.93 | 3.40 | 8.66 | 2.34 | 0.15 | 0.99 |
| Objectivity | Q3 | 6 | 254 | 385.43 | 3.08 | 6.67 | 2.36 | 0.33 | 0.99 |
| Objectivity | Q2 | 6 | 264 | 453.62 | 2.79 | 6.04 | 2.14 | 0.38 | 1.00 |
| Objectivity | Q1 | 6 | 279 | 557.23 | 2.87 | 6.63 | 2.12 | 0.41 | 1.00 |
| CIQ | Q4 | 24 | 265 | 403.51 | 3.66 | 20.39 | 2.94 | 0.29 | 0.99 |
| CIQ | Q3 | 24 | 254 | 675.34 | 2.80 | 11.25 | 2.43 | 0.49 | 1.00 |
| CIQ | Q2 | 24 | 264 | 796.64 | 2.50 | 10.60 | 2.15 | 0.55 | 1.00 |
| CIQ | Q1 | 24 | 279 | 882.81 | 2.45 | 11.76 | 2.05 | 0.56 | 1.00 |

Based on detailed ICC results for the hands-on task (see Table 1), we can argue in response to RQ2 that ICC has a positive correlation with motivation. The overall results for $\alpha$ reflect a high rate of scale reliability. As shown in Table 1, we reached the highest $\alpha$ for dimension completeness in all four quartile groups (Q4—Q1), which is also in line with the best results for ICC for that dimension.

In terms of ICC, participants attained the highest agreement levels for the dimension completeness, followed by accuracy, objectivity, and representation. For all four IQ dimensions, we obtained better results than existing studies [1, 4]. However, we must emphasize that our study is not a replication of studies from the literature. This study focuses on other sources under investigation (hints) and other aspects (motivation) that may influence inter-rater reliability.

We observe from our results that raters can consistently identify the quality of a hint if it leads to a problem solution; hence, participants could successfully rate completeness. It is also evident that raters can identify missing information and thus deduce completeness. The latter result is in line with the results of [1], who reported that the ICC for completeness was substantially higher than for other dimensions, although inter-rater agreement on this dimension was substantially lower in their research. Higher ICC may be obtained for completeness since people have a better understanding of this dimension than the other three dimensions. Participants determined the quality of a hint based on the possible hiding places left when they considered the hint. Since the task was straightforward, the participants succeeded in evaluating the quality of these hints and achieved better inter-rater agreement.

In terms of accuracy, we achieved a moderate (0.61) agreement. Compared to completeness (0.77), the lower result for accuracy may indicate that weight estimation is more challenging than locating an object. However, of all quality dimensions, accuracy gained the biggest increase in inter-rater reliability with a slightly increased motivation (Q4 and Q3).

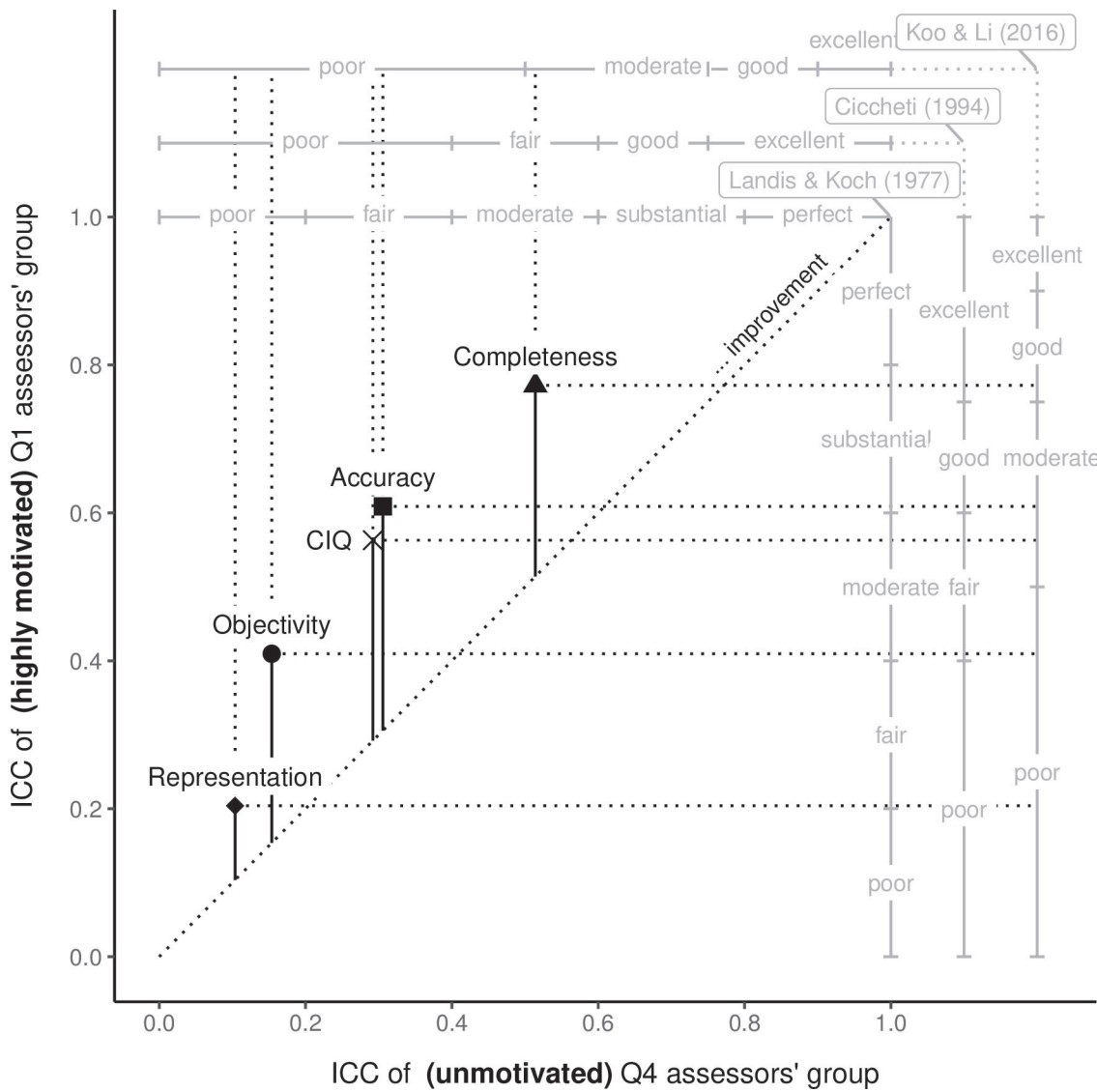

**Fig 4. Comparing inter-rater agreement for IQ dimensions in terms of motivation by assessors' groups.** The dual-scale data chart depicts the relationship between ICC values and various interpretations of inter-rater agreement. It compares the four IQ dimensions in terms of the extent to which motivation affected the increase in inter-rater agreement. The bottom x-axis denotes ICC for unmotivated users (Q4), while the left y-axis represents ICC values for highly motivated users (Q1). The scales on top of the x-axis and right of the y-axis denote various ICC interpretations for unmotivated (Q4) and motivated (Q1) users, respectively. The figure depicts all IQ dimensions and the mean value of the aforementioned dimensions. ICC values above the identity line (i.e. the dotted diagonal) represent an increase in ICC, while values below represent a decrease in ICC, when comparing unmotivated (Q4) and motivated (Q1) groups.

The measured ICC for the objectivity of motivated participants (Q1) is 0.41, noticeably lower than ICC for completeness (0.77) and accuracy (0.61). However, the result still demonstrates fair agreement. For non-motivated players (Q4), the ICC for objectivity was half that for accuracy and ICC noticeably increased with increased motivation. It is worth noting that the ICC trend for objectivity is very similar to that observed for completeness and accuracy. Not all participants recognized the relevant messages in the hints, and thus did not identify the quality of hints. The meaning of hints remained unclear because of the inclusion of scientific

terms, which offered assessors little clarification. As a result, inter-rater reliability was below expectations, though an upward trend was still present.

Participants attained the lowest agreement levels of all four dimensions for the representation dimension. Motivation had a positive impact on representation ICC, but not as much as in the case of other dimensions. Participants struggled with determining the consistency and hence the quality of the hint. As a direct consequence, inter-rater reliability was very low. We believe that this task was the most cognitively challenging of the four, and most participants chose not to invest much effort in solving it.

Based on [14–16] interpretations of ICC, we summarize the results of the study in Fig 4. Focusing on the groups of unmotivated participants (Q4) and the most motivated participants (Q1), the agreement levels were substantially higher for the motivated participants. Motivation increased the CIQ construct by 0.27 with the following interpretations (from Q4—to Q1): fair—moderate [14], poor—fair [15], and poor—moderate [16]. Comparing our results to findings in the literature [1, 4] further confirms that this study has achieved results with much greater ICC.

We found an increase in an agreement between the two groups of participants on all four IQ dimensions. We achieved the highest ICC agreement improvement for completeness, which increased from moderate (0.51) to good (0.77), according to the interpretation of [16]. The increase in ICC for completeness (0.26) was slightly below average CIQ (0.27), which still resulted in a superior result in an agreement for completeness. Interestingly, we observed the biggest improvement in ICC for the dimension of accuracy, with ICC increasing by as much as 0.30, improving from poor (0.31) to moderate (0.61) agreement, according to the interpretation of [16]. Motivation proved to be the key factor contributing to better accuracy assessment. The level of agreement of non-motivated participants for the dimensions objectivity (0.15) and representation (0.10) was poor. The introduction of motivation did not improve the quality of assessment enough to reach a moderate agreement for both dimensions (0.41 and 0.20 respectively). However, objectivity yielded an ICC increase slightly below average (0.26), indicating that motivation is a driving factor for this dimension. Nonetheless, objectivity remains difficult to evaluate consistently, even with motivated assessors. For representation, we observed the lowest agreement and ICC increase (0.10) between the four dimensions. Non-motivated participants achieved poor agreement levels (0.10). Motivation contributed to the rise in inter-rater agreement, but the result remained in the zone of poor agreement (0.20) according to the interpretation of [16].

## 4.3 Implications for research and practice

Our study supports the theoretical underpinning of IQ studies and confirms previous findings that IQ is a multidimensional construct that is difficult to measure. We also confirm that inter-rater agreement for different IQ dimensions can vary significantly.

Second, existing research has performed very poorly in assessing the measurability of IQ. According to most ICC interpretations, such results have very low measurability, so their interpretation is questionable. Our study builds on previous IQ-related research and extends it to alternative settings to demonstrate the significance of motivation in IQ assessment. Using gamified tasks to motivate assessors we were able to significantly improve the measurability of IQ (ICC). The correlation between points awarded in the gamified process and the inter-rater reliability agreement was positive for all four IQ dimensions and composite IQ (CIQ). The level of agreement achieved with the most motivated group of participants (Q1) was superior in comparison with results from related work. Future IQ measurement studies should take these results into account if they want interpretable results.

Third, the study extends previous research by introducing gamification to IQ assessment domain. It attempts to avoid rhetorical gamification by creating a renewed assessment process. The evidence reveals that gamification produced increased assessors' motivation leading to a better inter-rater agreement, consequently improving IQ assessment. That also confirms previous findings from [59, 62] stating that the gamification domain is immense and that researchers discover new application ares continuously.

Finally, the study attempts to motivate other researchers to replicate it in alternative settings to validate or complement our findings. Further studies should investigate additional factors that influence inter-rater reliability, such as heuristic principles used by participants, different sources of information, the size of the source under investigation, and what attributes of assessors affect results.

Information workers and researchers can benefit from our findings by creating IQ assessments in ways that take advantage of increased motivation. This study shows that we can achieve increased motivation by employing the concept of gamification, by including elements such as points, badges, and leaderboards.

Researchers studying the assessment of IQ should recognize motivation as a vital cue affecting IQ assessment. If applicable, they should consider including gamification in their studies. We conducted this study also with student participants. Our results demonstrate that gamification can be used successfully with students. Teachers creating gamified IQ tasks should consider improving the assessment process instead of adding gamification features to look like a gamified process.

Finally, the research provides insight into IQ and its dimensions for consumers of short online news. Many users are not aware of IQ dimensions and might start to consume contents in a more educated way.

## 4.4 Study validity

We performed activities both in the design phase and later in the data collection and analysis phases intended to increase the validity of our research.

To support internal validity, all participants involved in the experiment participated in the same gamified process, with equivalent study materials, questions, and the same method of obtaining data. The tool used in the experiment was intuitive and easy to use, so no special pre-test training was required for participation, although we performed an initial introduction to IQ dimensions to ensure participants understood the metrics being measured, as outlined in section 3.2. To minimize the instrumentation threat, we captured measured variables automatically and accurately. The participants were not aware of the research goal; they simply aimed to achieve the highest score within the gamified environment.

External validity requirements were addressed properly; our experimental setup represents a real-world situation and our test population has all knowledge expected of the general population. To maximize external validity, we followed the requirements outlined by [63]. As far as a generalization is concerned, the findings in [64] reveal a considerable similarity between many treatment effects obtained from the convenience and nationally representative population-based samples.

Concerning construct validity, participants were not subject to any pressure, and participation in the study was voluntary, which minimized mortality threat. In order to avoid unintentionally influencing the participants' behavior, there was no interaction between researchers and participants during the experiment or the study's goals. The problem domains in the gamified environment were selected to minimize any bias introduced by the familiarity of participants with given domains, which could have skewed the results in favor of some participants.

In terms of conclusion validity, we employed a robust measurement of inter-rater reliability agreement, ICC, to derive statistically correct conclusions based on the collected data. To compare the results with the findings in the literature, we included several ICC interpretations. We argue that the number of participants and the data collected were sufficient to draw reliable conclusions. We provide an explanation that a rater's motivation affects the measurability of IQ.

## 4.5 Limitations

Nevertheless, our study has some limitations that should be acknowledged. We wanted to include as large and as heterogeneous group of participants as possible, so we made the study available to the widest possible audience. Because our study did not have within a lab setup, we could not control all aspects of IQ assessment. For example, we were not able to assure that the participant completed the entire survey alone without assistance. However, by following the requirements outlined by [63] and iteratively improving the study in the study design phase, we believe our findings are relevant. As discussed in section 3.2 we also addressed this issue by preprocessing and cleaning of obtained data.

One limitation of this study is that people of the same age groups are not fully equally represented with a median value of 20 years old as presented in section 3.2. Research was available to the widest possible audience, so we had limited influence on the age of the participants. In would thus prove insightful to replicate the study where all age groups are equally represented. Although we found no statistically significant differences, it has been found in the literature that some demographic factors may affect the perceived benefits of gamification [65].

We should also be aware of the limitations that come from the ability of the participant to assess the quality of the object according to information quality [2]. Information quality assessment proved to be difficult. In their study, [1], showed that achieving agreement among assessors can be challenging.

For hints we used paragraph size documents instead of full size text documents. [4] found that shortening the text from full-size text to paragraph-size text does not affect the agreemenet level of information quality evaluations. However, future studies should thus consider using full-size text hints, which might lead to better user experience despite retaining IQ perception.

Finally, only one problem domain has been used in our study, as presented in section 3.1. Creating gamified content for additional domains requires lots of effort, especially defining game levels for evaluating specific IQ dimensions. Hence, our study should also be applied to other problem domains in future study replications.

## 5 Conclusion

Reaching consensus on IQ assessment is challenging, and the factors that drive successful estimation of IQ have not been fully explored. This study extends related work and confirms the effect of motivation as a driving factor for improved IQ assessment. It concludes that the employment of innovative gamified IQ assessment was effective, particularly for IQ dimensions that proved to be more reliable to consistent judgment in the literature. It increased participant engagement through the assessment content shortening and the inclusion of gamification features like points, levels, progress bars, and leader-board.

The level of agreement achieved with the most motivated group of participants (Q1) was superior in comparison with results from related work. Concerning the inter-rater agreement across the four IQ dimensions, we demonstrate that the relationship between individual IQ dimensions varies with motivation. With increasing motivation, the inter-rater agreement

consistently improved for the dimensions of objectivity, completeness, and accuracy. For the representation dimension, inter-rater reliability improved in the initial three quartiles.

Overall, gamification proved to be very useful in the field of IQ assessment. Thus, we strongly recommend that further IQ assessment studies control for the influence of motivation, and consider including a gamification approach. With the investigation of a different IQ source, foreknowledge might also be a key factor. Further studies could investigate the association between the amount of foreknowledge and inter-rater reliability.

## Author Contributions

**Conceptualization:** Marko Poženel, Dejan Lavbič.

**Formal analysis:** Marko Poženel.

**Methodology:** Dejan Lavbič.

**Project administration:** Dejan Lavbič.

**Supervision:** Dejan Lavbič.

**Validation:** Marko Poženel, Aljaž Zrnec.

**Writing – original draft:** Marko Poženel, Dejan Lavbič.

**Writing – review & editing:** Marko Poženel, Aljaž Zrnec, Dejan Lavbič.

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
