## [Decision Letter · Decision Letter 0]

20 Jun 2022

PONE-D-22-12464Measuring how motivation affects information quality assessment: a gamification approachPLOS ONE

Dear Dr. Lavbič,

Thank you for submitting your manuscript to PLOS ONE. After careful consideration, we feel that it has merit but does not fully meet PLOS ONE’s publication criteria as it currently stands. Therefore, we invite you to submit a revised version of the manuscript that addresses the points raised during the review process.

We look forward to receiving your revised manuscript.

Kind regards,

Ali Safaa Sadiq

Academic Editor

PLOS ONE

Journal Requirements:

Reviewers' comments:

Reviewer's Responses to Questions

**Comments to the Author**

1. Is the manuscript technically sound, and do the data support the conclusions?

Reviewer #1: Yes

Reviewer #2: Yes

2. Has the statistical analysis been performed appropriately and rigorously? 

Reviewer #1: Yes

Reviewer #2: Yes

3. Have the authors made all data underlying the findings in their manuscript fully available?

Reviewer #1: Yes

Reviewer #2: Yes

4. Is the manuscript presented in an intelligible fashion and written in standard English?

Reviewer #1: Yes

Reviewer #2: Yes

5. Review Comments to the Author

Reviewer #1: Introduction

The study deals with the important problem of poor information quality and the difficulties in assessing it. Leaving in an era of too much information and disinformation, this study is timely and very well focused i.e. Measuring the IQ of unstructured data ,especially with inter-rater agreement results using Interclass Correlation Coefficient (ICC) statistics (l.64).

The study focuses on motivation; however, motivation is not defined as a concept;

2 Related work

2.1 Assessing the quality of information

The discussion of what is quality of information; previous works on its various dimensions etc should come before the discussion of its assessment.

For example, it states on l. 117, R. Y.Wang and Strong (1996) defined a set of dimensions and a framework where dimensions are grouped into a hierarchical model of IQ aspects and their criteria.

First this citation is old; are there any more newer frameworks? Also, it mentions just one model, are there any more models? Or other IQ dimensions?

2.2 Gamification and motivation

As I noted above, motivation is not discussed thoroughly; there is a brief argument in this section however motivation needs to be properly discussed and defined. Gamification on the other hand is described in much more detail and its application is justified.

2.3 Problem statement and proposed solution

This section is well developed

3 Method

3.1 Evaluation mechanics

3.2 Participants

The study run from April 2015 to March 2020. Why so long? Typically, experiments took few weeks/months.

l. 255, the participants were 57.6% undergraduate students and 42.4% non-students. More demographics are needed; gender distribution, etc. selection criteria are not clear.

3.3 Measuring IQ dimensions: it is detailed and well presented

4 Results and discussion

Results are detailed. However, motivation is not clear how it was measured. Since motivation is pivotal to this study; motivation needs much more detailed presentation. For example, Table 1: ICC results in our research, shows Q depending on results; however, later on, it presents Q as motivation: e.g., p. 26 highly motivated users (Q1).

4.2 Discussion

There is a detail discussion but it looks confusing to see Q as motivation; in order to arrive at safe conclusions, motivation concept and measurements needs justification and better explanation.

4.3.1 Theoretical implications

Gamification could be a potential methodological contribution; I don’t see any theories discussed in the background study to understand how this study makes a theoretical contribution.

Recommendation

Major revision based on the comments above

Reviewer #2: The study is an important and interesting study that investigates how assessors' motivation affects information quality and how more motivated assessors can improve inter-rater agreement among different assessors. The paper is well-written. However, there are minor and major comments that need to be addressed.

Major Comments

1. Missing citations. Many statements seem very arbitrary and require citations. Below are a few examples of where citations are required:

Introduction, "Making the best possible decisions requires information of the highest quality. As the amount of information available grows, it becomes increasingly difficult to distinguish quality from questionable information. The problem of poor information quality can weaken our decision processes, so we need more reliable measures and new techniques to assess the quality of information. Unfortunately, such assessment can itself be very demanding."

Introduction, "The literature has widely adopted a multidimensional view of IQ to support more effortless management of its complexity."

Intrdouction, "Regardless of the ICC interpretation used, the values reported in recent studies are poor or at best moderate. This demonstrates that reaching consensus among various raters is difficult when measuring IQ."

Gamification and motivation, "Gamification (Gameful design) is a concept where we use game-like elements in various systems to increase user participation, motivation, improve engagement, or to retain users continue using the system."

There are many similar problems like the ones I listed above. Please check statements that require citations.

2. For grouping motivating participants from less motivating participants, what factors have been considered in addition to the player's obtained scores? Please consider mentioning more about it.

3. What are the benefits and rationals behind using short article length for your study? Please consider mentioning more about it.

4. Motivation is essential, as discussed in the article. It could be of interest to present more about intrinsic and extrinsic motivation and how both can be considered in a gamified process or apps. E.g., how can different game elements satisfy players' intrinsic and extrinsic motivations?

5. Methods, Did the gamified process used in experiments designed and developed by the authors? Why was this gamified process and design chosen for the study instead of alternatives?

6. Limitations of the study should be acknowledged?

7. Completeness, accuracy, representation, and objectivity dimensions of IQ should be introduced and defined shortly. After reviewing the paper twice, I cannot get an exact definition of these terms. Your definition may be different from what the paper's readers and I think.

Minor Comments

8. Abstract/ purpose: In the following statement, I think the ICC is the abbreviation of Interclass Correlation Coefficient, not inter-rater agreement. Please check and revise it.

"Existing research on the measurability of information quality (IQ) has delivered poor results

and demonstrated low inter-rater agreement (ICC) in evaluating IQ dimensions."

9. Gamification and motivation: In the following sentence, I suggest referring Deterding et al., who defined this definition of gamification. I recommend also referring to Pelling in this section, who originally coined the concept of gamification:

"In the literature, gamification is often defined as the use of game design elements in non-game contexts (Bovermann and Bastiaens 2020)."

Deterding S, Dixon D, Khaled R, Nacke L. From Game Design Elements to Gamefulness: Defining "Gamification". 2011 Presented at: MindTrek '11: 15th International Academic MindTrek Conference: Envisioning Future Media Environments; September 28-30, 2011; Tampere, Finland p. 9-15. [doi: 10.1145/2181037.2181040]

Pelling, N. (2011, August 9). The (short) prehistory of "gamification"…. Funding Startups (& Other Impossibilities). https://nanodome.wordpress.com/2011/08/09/the-short-prehistory-of-gamification/

10. Gamification and motivation, In the following sentence, replacing "In-game gamification features" with "game elements" is better.

"In-game gamification features include points, levels, choice elements, progress bars, and leaderboards."

11. Gamification and motivation, In the following sentence, for each filed like sports, one or more related studies should be cited, not Hamari 2017

“It is used in application fields like sports, health, sustainability, education, marketing, and business (Hamari 2017).”

12. Gamification and motivation, in the following statements, authors should use phrases such as “as far as we now” or “to our knowledge”. Authors can not make such a claim unless they have strong evidence and references to support it.

“Previously, gamification has not been used to improve IQ assessment and inter-rater agreement.”

6. PLOS authors have the option to publish the peer review history of their article (what does this mean?). If published, this will include your full peer review and any attached files.

Reviewer #1: No

Reviewer #2: **Yes: **Ali Khaleghi

---

## [Author Response · Author response to Decision Letter 0]

17 Jul 2022

A comprehensive response to reviewers is included in a separate file.

---

## [Decision Letter · Decision Letter 1]

5 Sep 2022

Measuring how motivation affects information quality assessment: a gamification approach

PONE-D-22-12464R1

Dear Dr. Lavbič,

We’re pleased to inform you that your manuscript has been judged scientifically suitable for publication and will be formally accepted for publication once it meets all outstanding technical requirements.

Kind regards,

Ali Safaa Sadiq

Academic Editor

PLOS ONE

Additional Editor Comments (optional):

Authors could address all the given comments and I am happy to proceed with the possible publication for their manuscript.

Reviewers' comments:

Reviewer's Responses to Questions

**Comments to the Author**

1. If the authors have adequately addressed your comments raised in a previous round of review and you feel that this manuscript is now acceptable for publication, you may indicate that here to bypass the “Comments to the Author” section, enter your conflict of interest statement in the “Confidential to Editor” section, and submit your "Accept" recommendation.

Reviewer #1: All comments have been addressed

2. Is the manuscript technically sound, and do the data support the conclusions?

Reviewer #1: Yes

3. Has the statistical analysis been performed appropriately and rigorously? 

Reviewer #1: Yes

4. Have the authors made all data underlying the findings in their manuscript fully available?

Reviewer #1: Yes

5. Is the manuscript presented in an intelligible fashion and written in standard English?

Reviewer #1: Yes

6. Review Comments to the Author

Reviewer #1: (No Response)

7. PLOS authors have the option to publish the peer review history of their article (what does this mean?). If published, this will include your full peer review and any attached files.

Reviewer #1: No

---

## [Editor Report · Acceptance letter]

18 Oct 2022

PONE-D-22-12464R1 

Measuring how motivation affects information quality assessment: a gamification approach 

Dear Dr. Lavbič:

I'm pleased to inform you that your manuscript has been deemed suitable for publication in PLOS ONE. Congratulations! Your manuscript is now with our production department. 

Kind regards, 

on behalf of

Dr. Ali Safaa Sadiq 

Academic Editor

PLOS ONE